# Frequency-specific attentional mechanisms phasically modulate the influence of distractors on task performance

Zach V. Redding*, Yun Ding, Ian C. Fiebelkorn*

Department of Neuroscience and Ernest J. Del Monte Institute for Neuroscience, University of Rochester, Rochester, New York, United States of America

* Zachary_Redding@URMC.Rochester.edu (ZVR); Ian_Fiebelkorn@URMC.Rochester.edu (ICF)

## Abstract

The Rhythmic Theory of Attention proposes that visual spatial attention is characterized by alternating states that promote either sampling at the present focus of attention or a higher likelihood of shifting attentional resources to another location. While theta-rhythmically (4–8 Hz) occurring windows of opportunity for shifting attentional resources might provide cognitive flexibility, these windows might also make us more susceptible to distractors. Here, we used EEG in humans to test how frequency-specific neural activity phasically influences behavioral performance and visual processing when high-contrast distractors co-occur with low-contrast targets. For trials with and without distractors, perceptual sensitivity at the cued target location depended on pre-stimulus theta phase (~7 Hz) recorded at central electrodes. For trials with distractors, there was a greater increase in false alarm rates at the same theta phase associated with lower hit rates (i.e., during the proposed "shifting state"), confirming theta-rhythmically occurring windows of increased susceptibility to distractors. In addition to these phase–behavior effects at central electrodes, we observed phase–behavior effects at frontocentral and occipital electrodes that (i) only occurred on trials with distractors, (ii) peaked in the alpha-frequency range (~9–10 Hz), and (iii) were strongest at occipital electrodes that were contralateral to distractors. Alpha phase at these electrodes was also associated with fluctuations in the amplitude of distractor-evoked visual responses, consistent with an alpha-mediated gating of distractors. The present findings thus provide evidence for distinct theta- and alpha-mediated mechanisms of spatial attention that phasically modulate the influence of distractors on task performance.

## Introduction

The brain has limited processing resources. It therefore uses a collection of filtering mechanisms to enhance the processing of behaviorally important information and

**Data availability statement:** All data files and code used for the main and supplementary analyses are available from The Open Science Framework at https://doi.org/10.17605/OSF.IO/V24GU.

**Funding:** This study received funding from the National Institutes of Health, (https://ror.org/01cwqze88, R01EY033726 to ICF), the National Science Foundation, (https://ror.org/021nxhr62, 2120539 to ICF) and the Searle Scholars Program (https://searlescholars.org/, to ICF). The funders had no role in study design, data collection and analysis, decision to publish, or preparation of the manuscript.

**Competing interests:** The authors have declared that no competing interests exist.

**Abbreviations:** DISS, global dissimilarity measures; dva, degrees of visual angle; FAR, false alarm rate; HR, hit rate.

suppress the processing of distracting information. Spatial attention specifically refers to the filtering mechanisms through which the brain preferentially processes information at behaviorally important locations in space [1]. The deployment of spatial attention is associated with changes in neural activity that improve behavioral performance [2–4]. These neural and behavioral effects, however, are not sustained during attentional deployment. Growing evidence instead indicates that attention-related effects fluctuate over time, on a theta-rhythmic timescale (i.e., at 4–8 Hz) [5–13]. Fluctuations in attention-related effects have been linked to theta-rhythmic neural activity in the large-scale network that directs spatial attention and exploratory movements (i.e., the "Attention Network") [14–18].

The Rhythmic Theory of Attention proposes that the waxing and waning of attentional effects reflects alternating states associated with either sampling at the present focus of attention (i.e., a "sampling state") or a higher likelihood of shifting attentional resources to another location (i.e., a "shifting state") [4,7,11,19,20]. In support of this "shifting state," exploratory behaviors, such as eye movement in primates, have similarly been linked to theta-rhythmic neural activity [21–27]. Frequently occurring windows of opportunity for shifting attentional resources could prevent us from becoming overly focused on any single location in space, thereby providing critical cognitive flexibility. On the other hand, rhythmically occurring shifting states could make us more susceptible to distracting information (i.e., behaviorally irrelevant information). Previous auditory research, for example, has demonstrated that the vulnerability of working memory to distractors fluctuates at low frequencies [28,29]. Here, we tested whether frequency-specific neural activity phasically influences susceptibility to distractors during attention-related sampling of the visual environment, specifically when spatially predictable, high-contrast distractors co-occur with spatially predictable, low-contrast targets.

In addition to hypothesized links between theta-band activity and distractor susceptibility, alpha-band activity (9–14 Hz) has been repeatedly associated with the suppression of sensory processing [30,31]. Higher alpha power typically occurs within neural populations that are processing behaviorally irrelevant sensory information [32–34] and is also associated with lower cortical excitability [35–38]. For example, Haegens and colleagues [35] demonstrated that higher alpha power was associated with lower spike rates. Despite these often-observed relationships between alpha-band activity and the suppression of sensory processing, research examining whether alpha-mediated mechanisms of spatial attention can be actively deployed to suppress spatially predictable distractors has provided mixed results [39,40]. We therefore also tested whether pre-stimulus alpha-band activity—like pre-stimulus theta-band activity—modulates behavioral performance and visual processing when spatially predictable, high-contrast distractors co-occur with spatially predictable, low-contrast targets. The present findings provide evidence for distinct theta- and alpha-mediated mechanisms of spatial attention that both phasically modulate the influence of distractors on task performance. Moreover, these findings confirm a key prediction of the Rhythmic Theory of Attention: despite being behaviorally disadvantageous, there are theta-rhythmically occurring windows of increased susceptibility to distractors.

## Results

### Behavioral effects

Here, our primary goal was to test whether oscillatory mechanisms of spatial attention influence the effects of a distractor on visual-target detection. To do this, we presented spatially predictable targets with spatially predictable distractors (Fig 1), using spatial cues with 70% validity. Prior to measuring the influence of oscillatory mechanisms on behavioral performance, we tested (i) whether the distractor interfered with visual-target detection [41] and (ii) whether participants utilized the spatially informative target and distractor cues. Fig 2 shows that the presence of a distractor interfered with visual-target detection at cued target locations (relative to trials without a distractor), both reducing HR [$t_{(31)}$ = 4.163, $p < 0.001$, $d = 1.296$] and increasing FAR [$t_{(31)}$ = 2.954, $p = 0.006$, $d = 0.482$]. Combining HR and FAR, the presence of a distractor also reduced d′ [$t_{(31)}$ = 7.281, $p < 0.001$, $d = 0.868$], confirming that distractors were associated with reduced perceptual sensitivity at cued target locations (for individual scores see S1 Data).

We next tested whether spatially informative target and distractor cues were associated with better visual-target detection, relative to invalidly cued trials. Fig 3 illustrates both target- and distractor-cueing effects. Here, HR was significantly increased for validly cued targets, relative to invalidly cued targets [$t_{(31)}$ = 7.384, $p < 0.001$, $d = 1.79$], but there was no significant effect of target cueing on FAR [$t_{(31)}$ = 1.037, $p = 0.308$, $d = 0.236$]. Combining HR and FAR, d′ was significantly higher at cued target locations, relative to non-cued target locations [$t_{(31)}$ = 4.461, $p < 0.001$, $d = 0.818$] (Fig 3A–3C; for individual scores see S2 Data). Distractor cues were associated with a similar pattern of results. That is, HR was significantly increased when there was a validly cued distractor, relative to when there was an invalidly cued distractor [$t_{(31)}$ = 4.304, $p < 0.001$, $d = 0.753$], but there was no significant effect of distractor cueing on FAR [$t_{(31)}$ = 1.167, $p = 0.252$, $d = 0.128$].

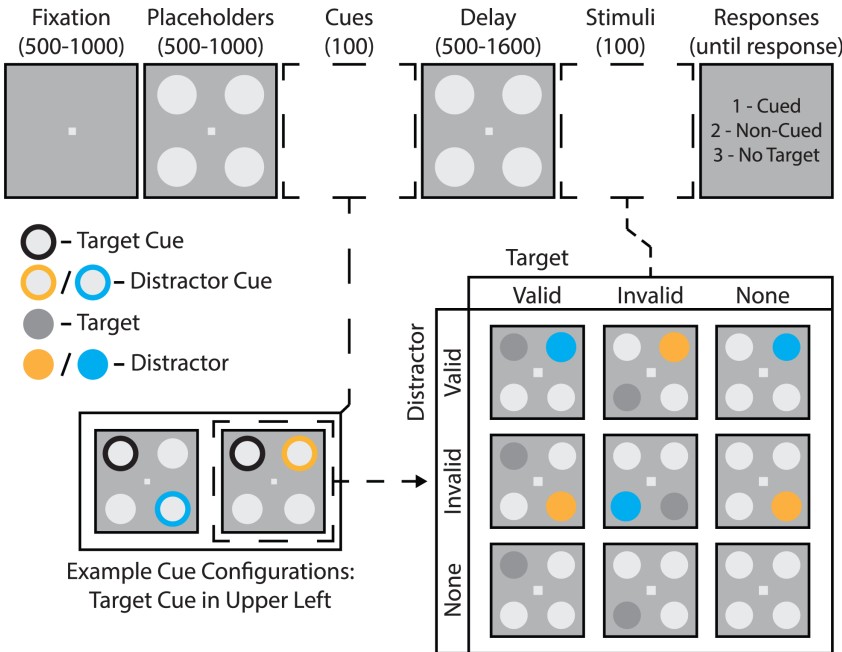

**Fig 1. Experimental task.** Trials began with variable fixation (500–1,000 ms) and placeholder (500–1,000 ms) intervals, followed by two spatial cues (100 ms) on opposite sides of the visual field to indicate the likely locations of both a subsequent near-threshold target and a salient distractor (100 ms). Cue validity was 70% for both cue types. Targets and distractors were presented after a variable delay (500–1,600 ms). Stimulus displays could include (i) both a target and a distractor, (ii) a target only, (iii) a distractor only, or (iv) neither a target nor a distractor. The number pad on a computer keyboard was used to indicate the presence of a target at the cued location, a target at a non-cued location, or no target.

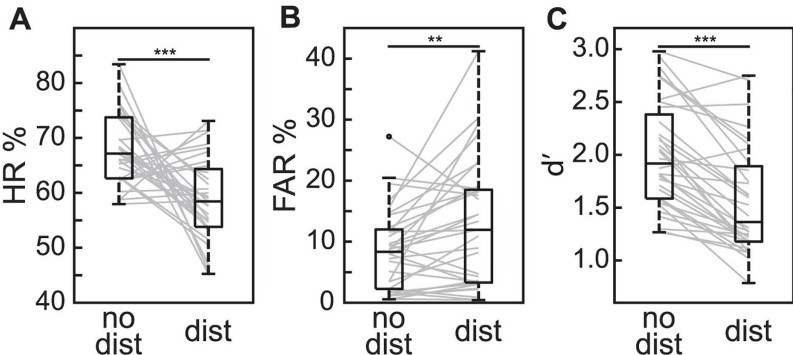

**Fig 2. Distractors impaired task performance. A**, HR for trials with no distractor and trials with a distractor. The line within each box represents the median, while the boundaries of each box represent the upper and lower quartiles, respectively. Lines outside the box represent maximum and minimum data points excluding outliers. Outliers were >1.5 times the interquartile range outside of upper and lower quartiles and are denoted with a dot. Gray lines indicate the change across conditions for individual subjects. **B**, As in **A**, but for FAR. **C**, As in **A**, but for d′. ** $p < 0.01$; *** $p < 0.001$. See S1 Data for individual scores.

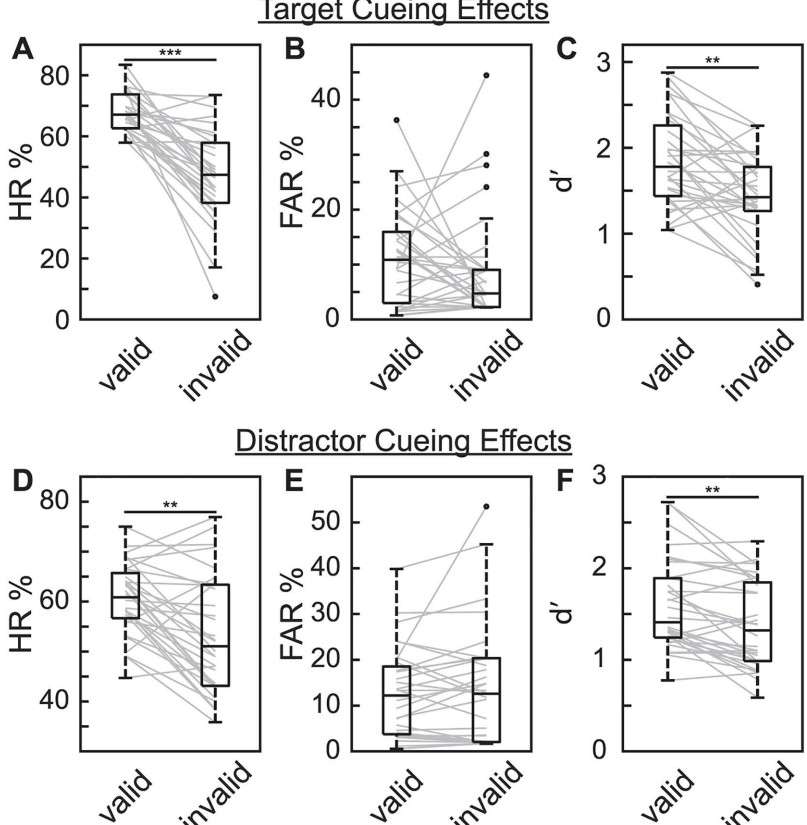

**Fig 3. Spatially informative target cues and distractor cues improved task performance. A**, HR for trials with valid targets and trials with invalid targets. The line within each box represents the median, while the boundaries of each box represent the upper and lower quartiles, respectively. Lines outside the box represent maximum and minimum data points excluding outliers. Outliers were >1.5 times the interquartile range outside of upper and lower quartiles and are denoted with a dot. Gray lines indicate the change across conditions for individual subjects. **B**, FAR for responses at valid (i.e., cued) locations and invalid (non-cued) locations. **C**, d′ for valid/cued locations and invalid/non-cued locations. **D**, HR for trials with valid distractors and trials with invalid distractors. **E**, As in **D**, but for FAR. **F**, As in **D**, but for d′. ** $p < 0.01$; *** $p < 0.001$. See S2 Data for individual scores.

Combining HR and FAR, d′ at the cued *target* location was significantly higher when there was a validly cued distractor, relative to when there was an invalidly cued distractor [$t_{(31)}$ = 4.065, $p < 0.001$, $d = 0.455$] (Fig 3D–3F; for individual scores see S2 Data). These behavioral results confirm *that* participants utilized the spatially informative target and distractor cues to both enhance target processing at likely target locations and suppress distractor processing at likely distractor locations.

## ERP evidence for cue-related suppression of distractor processing

After confirming behavioral evidence of cue-related distractor suppression, we tested for neurophysiological evidence of such suppression. Fig 4A depicts grand-averaged ERP waveforms in response to either validly or invalidly cued distractors that were presented in the contralateral hemifield. All analyses of distractor-evoked responses used occipital electrodes A10/B7 (corresponding to PO7/PO8), which were selected a priori based on past work on distractor-evoked visual responses [42–44], as well as past work on other attention-related effects associated with visual responses, such as the N2pc component [45]. There is a prominent P1 component, consistent with processing in extrastriate areas [46]. Fig 4B depicts the difference wave associated with these distractor-evoked potentials (i.e., invalid minus valid). Cluster-based permutation tests comparing the visual responses to validly and invalidly cued distractors revealed a significant cluster from 136.5 to 152.1 ms after distractor onset ($p = 0.016$). The topography of the difference wave during this temporal window is consistent with a suppressive effect on sensory processing in occipital cortex (Fig 4C). Here, we flipped the topography from trials when the distractor occurred on the right side of the visual field, such that the right side of the present topography represents the visual response at electrodes that were contralateral to distractors (i.e., regardless of whether the distractor was presented to the right or to the left of central fixation) and the left side of the present topography represents the visual response at electrodes that were ipsilateral to distractors. These neurophysiological results provide further evidence that participants utilized the spatially informative distractor cue to suppress distractor processing. The low-contrast target did not elicit clear evoked responses (S1 Fig), so we were unable to test for neurophysiological evidence of cue-related target enhancement.

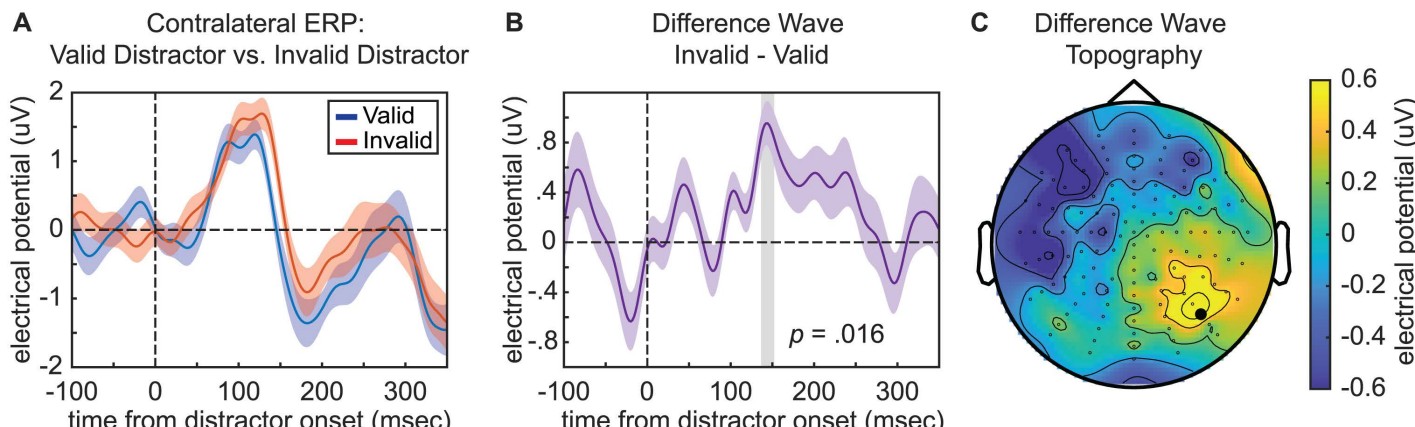

**Fig 4. ERP evidence for suppression of distractors appearing at cued locations. A**, Grand average ERPs from occipital channels contralateral to valid distractors (blue), invalid distractors (red). **B**, Grand average difference wave after subtracting valid-distractor responses from invalid distractor responses. The time window of significant difference between conditions is indicated with a gray bar (136.5–152.1 ms). All error bars indicate ±1 SEM. **C**, Topography of the difference wave averaged over the significant time window. The electrode used for ERP analyses is depicted with a bold dot. The topography is configured as if all stimuli appeared on the left side of the visual field (i.e., channels were flipped for trials with stimulus on the right).

## Distinct oscillatory mechanisms phasically modulate the influence of distractors on task performance

We next tested our primary research question, measuring the link between the pre-stimulus phase of frequency-specific neural activity and visual-target detection at the cued target location (Fig 5). We specifically measured these phase–behavior relationships with and without a validly cued distractor. Fig 6 displays significant phase–behavior relationships—based on cluster-based statistics (see Materials and methods)—across behavioral measures and experimental conditions (Fig 6A). For trials with and without a distractor, the phase of theta-band activity significantly modulated HR and d′ (Fig 6A). For trials without a distractor, these effects were strongest at ~7 Hz, with topographies of the d′ effects revealing a central cluster and broad effects across bilateral frontal channels, with only frontal channels contralateral to the distractor cue reaching statistical significance (Fig 7). FAR, in comparison, was only linked to theta phase on trials with a distractor (Fig 6A). Notably, the phase associated with the lowest HR was also associated with the highest distractor-related increase in FAR (Fig 8B). These results are consistent with the proposal that both visual target detection and distractor susceptibility fluctuate as a function of attention-related, theta-rhythmic sampling [4].

For trials with a distractor, all behavioral measures (i.e., HR, FAR, and d′) were also linked to the phase of higher frequency oscillations in the alpha band, with the strongest effects at ~9–10 Hz (Figs 6A, 9A, 9B, and S2). Topographies of the d′ effects revealed alpha-specific frontocentral and occipital clusters, with the occipital effects being stronger at electrodes that were contralateral to the distractor cue (Fig 7). As with the ERP analyses, we flipped the topographies from trials when the distractor cue occurred on the right side of the visual field, such that the right side of topographies represents phase–behavior relationships at electrodes that were contralateral to distractor cues and the left side represents phase–behavior relationships at electrodes that were ipsilateral to distractors cues. The alpha phase associated with better behavioral performance at the frontocentral cluster (Fig 9B) is opposite from the alpha phase associated with better behavioral performance at the occipital cluster (S2 Fig). These antiphase results could reflect opposite ends of an

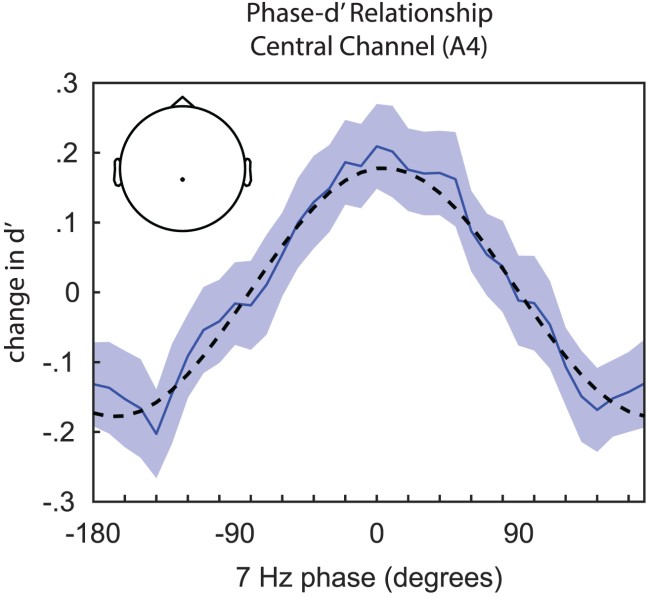

**Fig 5. Example of phase–behavior analysis.** Normalized behavioral measures are plotted as a function of the pre-stimulus phase of ongoing EEG oscillations. This example depicts the relationship between 7 Hz phase and d′ (solid blue line). Error bars indicate ±1 SEM. The amplitude of a fitted, one-cycle sine wave (dashed line) indicates the strength of the phase–behavior relationship. Inset depicts the electrode associated with this phase–behavior function.

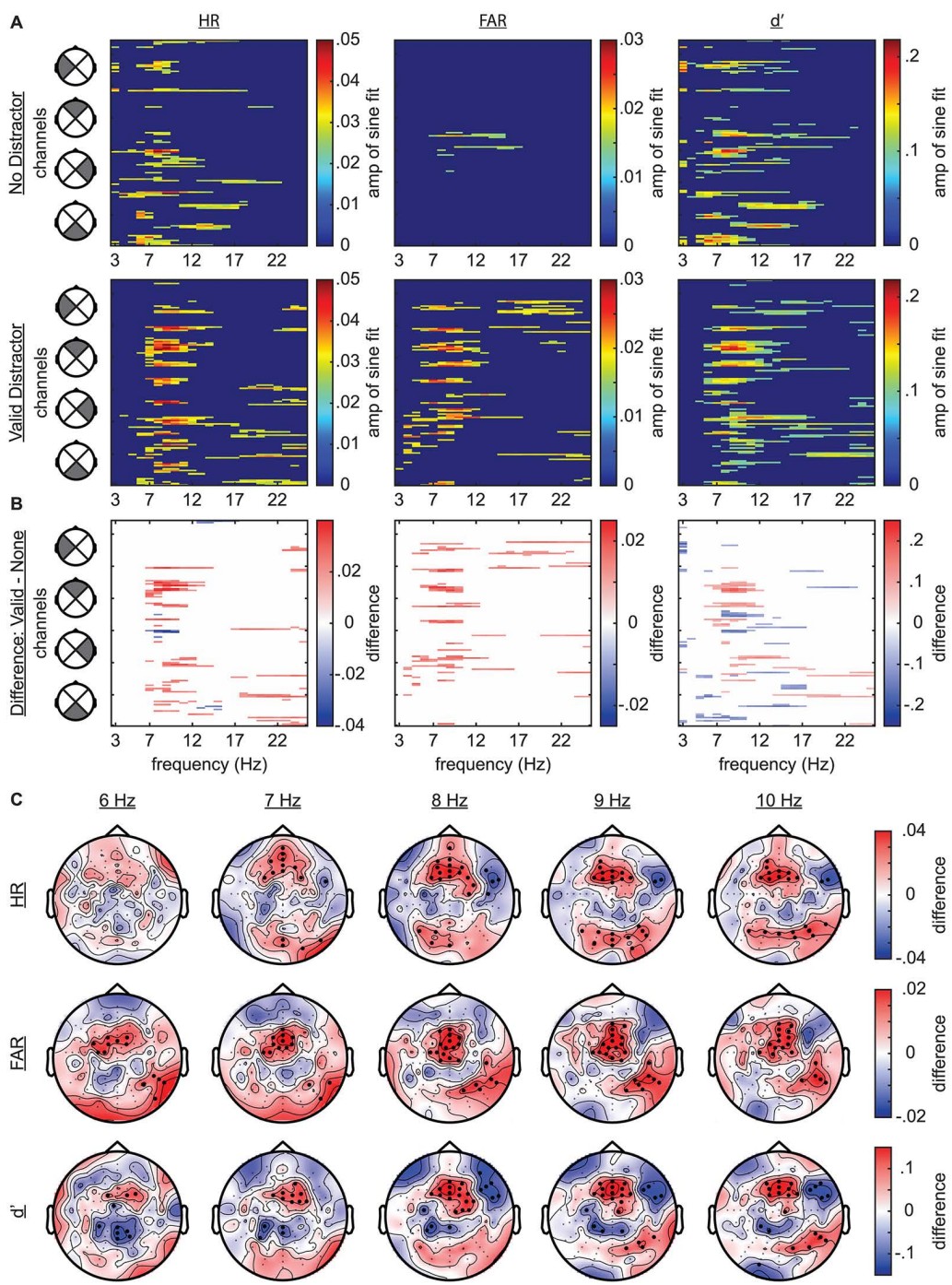

**Fig 6. Phase–behavior relationships. A**, Plots depicting the strength of statistically significant ($p < 0.05$) phase–behavior relationships as a function of frequency (3–25 Hz) across all 128 channels. Statistically insignificant results were set to zero. The first column depicts phase-HR relationships, the second depicts phase-FAR relationships, and the third depicts phase-d′ relationships. The top row shows effects for trials with no distractor and the bottom row shows these effects for trials with a valid distractor. **B**, Plots depicting differences in phase–behavior coupling between conditions (valid distractor minus no distractor). Statistically insignificant results were set to zero. **C**, Topographies depicting the differences plotted in **B** for frequencies from 6 to 10 Hz (columns). The top row depicts phase-HR differences, the middle row depicts phase-FAR differences, and the bottom row depicts phase-d′ differences. Channels with a significant effect at a given frequency are indicated by bold dots. The topographies are configured as if all stimuli appeared on the left side of the visual field (i.e., channels were flipped for trials with stimulus on the right).

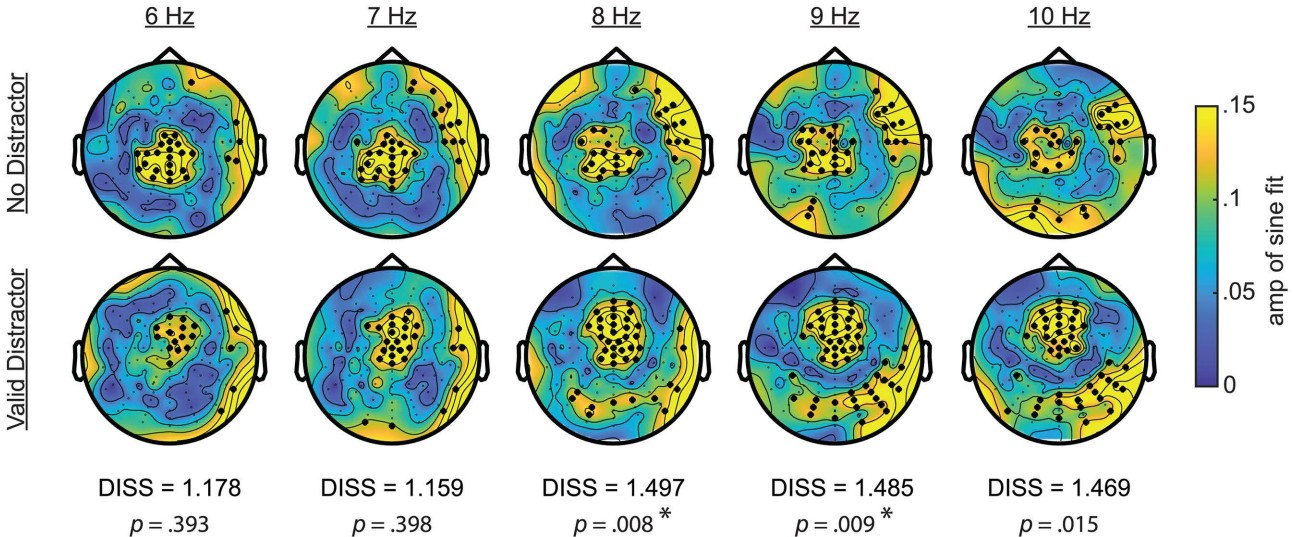

**Fig 7. Topographic analyses.** Scalp topographies show the strength of phase-d′ relationships for frequencies from 6 to 10 Hz (columns). The top row shows effects for trials with no distractor and the bottom row shows effects for trials with a valid distractor. Channels with a significant effect at a given frequency are indicated by bold dots. Global dissimilarity measures (DISS) and *p*-values for tests comparing topographies across the no-distractor and valid-distractor conditions are listed below the respective frequency, with significant values after correcting for multiple comparisons (*p* < 0.05) indicated with an asterisk. The topographies are configured as if all stimuli appeared on the left side of the visual field (i.e., channels were flipped for trials with stimulus on the right).

alpha-related dipole, or perhaps align with a recent proposal that top-down distractor suppression is mediated by anti-phase alpha-band activity in distinct sensory and downstream detection areas [47].

Previous work has similarly linked fluctuations in the likelihood of microsaccades—fixational eye movement of < 2 degrees of visual angle (dva)—with frequency-specific phase [22]. As the direction of microsaccades are associated with attentional deployment and attentional shifts [48,49], we re-calculated phase–d′ relationships after excluding trials with microsaccades (7.5% of trials) that occurred prior to stimulus presentation (i.e., prior to the presentation of targets and/or distractors). As in previous work [14], this control analysis generally demonstrated the same pattern of phase–behavior relationships after the exclusion of trials with microsaccades (S3 Fig), confirming that phase–behavior relationships are not attributable to microsaccades.

To further test how phase–behavior relationships differed between trials with and without distractors, we used (i) cluster-based permutation tests to compare phase–behavior relationships across trials with and without a distractor and (ii) global dissimilarity measures (DISS) to compare the phase–behavior topographies observed across these same conditions [50]. Cluster-based permutation tests revealed that while central theta effects (d′) were apparent for both conditions, these effects were stronger for trials without distractors (Fig 6B and 6C). Trials with a distractor, in contrast, exhibited significantly greater coupling between alpha phase and all behavioral measures (HR, FAR, and d′) at frontocentral electrodes and at posterior electrodes that were contralateral to the distractor (Fig 6B and 6C). Statistical comparisons of the topographies for phase-d′ effects (distractor versus no-distractor trials), from 6 to 10 Hz (Fig 7; $\alpha = 0.01$), revealed no differences at 6 Hz (DISS = 1.178; *p* = 0.393) and 7 Hz (DISS = 1.159; *p* = 0.398) and significant differences at 8 Hz (DISS = 1.497; *p* = 0.008) and 9 Hz (DISS = 1.485; *p* = 0.009). The test for 10 Hz (DISS = 1.469; *p* = 0.015) did not survive the correction for multiple comparisons. Combined, these analyses reveal (i) theta-dependent effects on trials with and without a distractor and (ii) alpha-dependent effects that only occurred on trials with a distractor.

Finally, we tested whether pre-stimulus theta and alpha phase were associated with fluctuations in distractor-evoked visual responses. Here, we used broadband power (from 1 to 55 Hz), rather than grand-averaged ERPs (see Materials and methods). Using the channel with the strongest relationship between pre-stimulus phase and d′ on trials with a distractor (i.e., channel C23 for 7 Hz and channel C22 for 9 Hz), we binned trials by phase (±60° of the midpoint of bins with the best and worst behavioral performance) to create "good" and "bad" phase conditions (Figs 8C and 9C). As with our previous analysis (Fig 4), we calculated distractor-evoked responses at occipital electrodes A10/B7 (corresponding to PO7/PO8). While there was no effect of theta phase (Fig 8D) on distractor-evoked responses, there was a significant effect of alpha phase on distractor-evoked responses (Fig 9D). That is, the response to valid distractors was weaker when presented during the "good" alpha phase relative to the "bad" alpha phase, with a cluster-based permutation approach (see Materials and methods) revealing a significant cluster from 163.8 to 218.5 ms after distractor onset ($p = 0.001$). We also used a cluster-based permutation test to directly compare the influence of theta and alpha phase on distractor-evoked responses, revealing a significantly greater effect of alpha phase ($p = 0.024$) from 159.9 to 206.8 ms relative to distractor onset (S4 Fig). These results are consistent with an alpha-mediated gating of distractor processing [30].

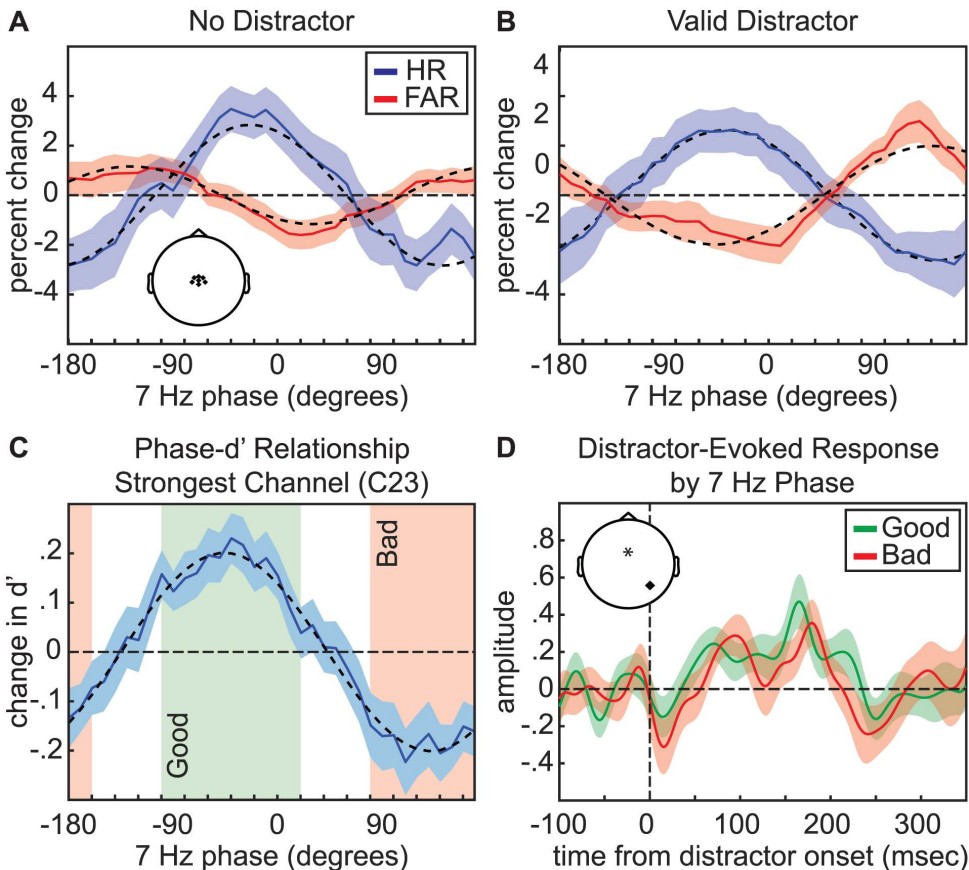

**Fig 8. Task performance depends on pre-stimulus theta phase. A**, Phase-HR (solid blue line) and phase-FAR (solid red line) relationships for 7-Hz phase using trials with no distractor. These effects were averaged across central channels that were significant for both conditions (depicted in inset). **B**, As in **A**, but for trials with a valid distractor. **C**, Phase-d′ relationship for the channel (C23) demonstrating the strongest effect on trials with a valid distractor (solid blue line). This function was used to determine "good" (green box) and "bad" (red box) phase bins. **D**, Distractor-evoked broadband (1–55 Hz) response as a function of central 7-Hz phase ("goo" vs. "bad"). Distractor-evoked broadband responses were calculated for electrode A10 or electrode B7, depending on whether the distractor occurred on either the left or right of central fixation. Inset depicts the locations of the channel used for phase measurements (indicated by an asterisk) and the channel used for broadband responses (indicated with a dot). All error bars indicate ±1 SEM.

## Occipital alpha power does not influence suppression of distractors at expected locations

While alpha-band activity has been repeatedly linked to the suppression of sensory processing, evidence for a relationship between alpha-band activity and distractor suppression has been mixed [40]. In the previous analysis, the results demonstrated a significant relationship between pre-distractor alpha phase and subsequent behavioral and neurophysiological responses (Figs 6 and 9). In addition to frontal electrodes, these effects occurred at electrodes contralateral to the presentation of distractors. This is consistent with lateralized alpha-band activity during the deployment of spatial attention [33]. Given ongoing debates regarding whether alpha-associated suppression functions through phasic modulation and/or power modulation, we tested whether pre-stimulus alpha power was also linked to behavioral performance on trials with distractors. We specifically tested whether alpha-power lateralization (i.e., alpha power at electrodes contralateral to the distractor cue relative to alpha power at electrodes ipsilateral to the distractor cue) was linked to behavioral performance. Power spectra for five subjects did not display obvious alpha-band peaks. These subjects were therefore excluded from the present analyses ($n = 27$). The mean peak alpha frequency (across subjects) was 10.95 Hz (Fig 10A). The topography

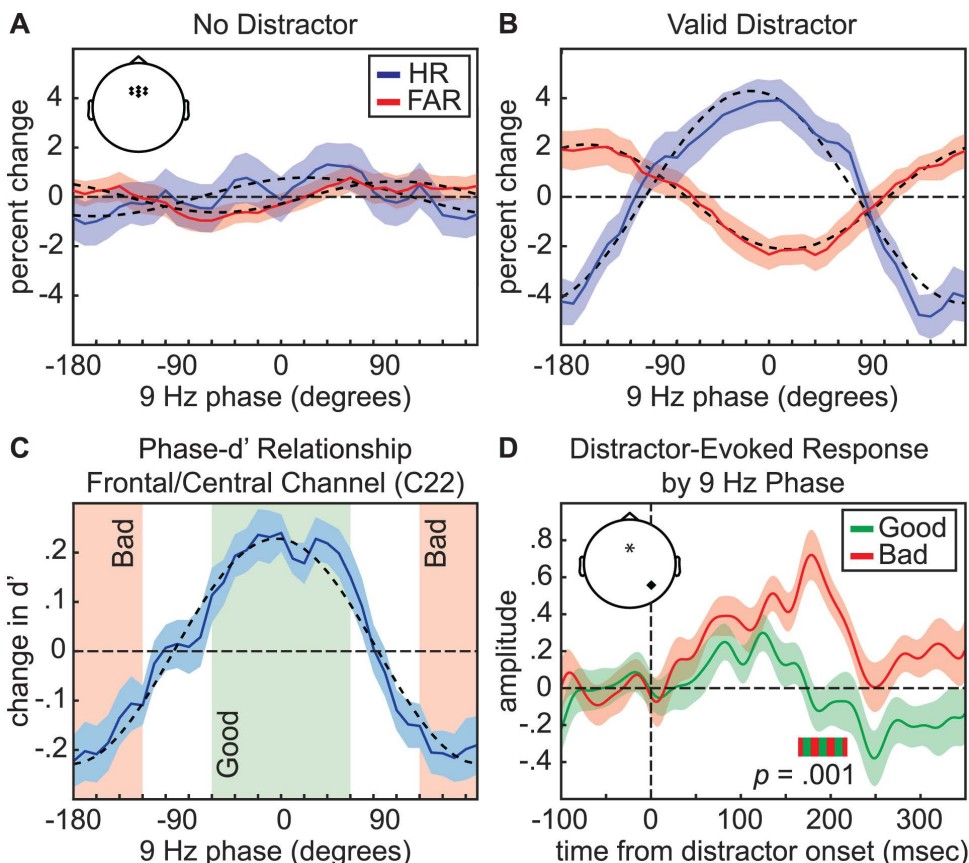

**Fig 9. Distractor suppression depends on pre-stimulus alpha phase. A**, Phase-HR (solid blue line) and phase-FAR (solid red line) relationships for 9–Hz phase using trials with no distractor. These effects were averaged across frontocentral channels that were significant for trials with a valid distractor (depicted in inset). **B**, Same as **A**, but for trials with a valid distractor. **C**, Phase-d′ relationship for the channel (C22) demonstrating the strongest effect on trials with a valid distractor (solid blue line). This function was used to determine "good" (green box) and "bad" (red box) phase bins. **D**, Distractor-evoked broadband (1–55 Hz) response as a function of frontocentral 9-Hz phase ("good" vs. "bad"). Distractor-evoked broadband responses were calculated for electrode A10 or electrode B7, depending on whether the distractor occurred on either the left or right of central fixation. Inset depicts the locations of the channel used for phase measurements (indicated by an asterisk) and channel used for broadband responses (indicated with a dot). All error bars indicate ±1 SEM. Horizontal green and red bar indicates the time window for the significant difference between conditions (163.8–218.5 ms).

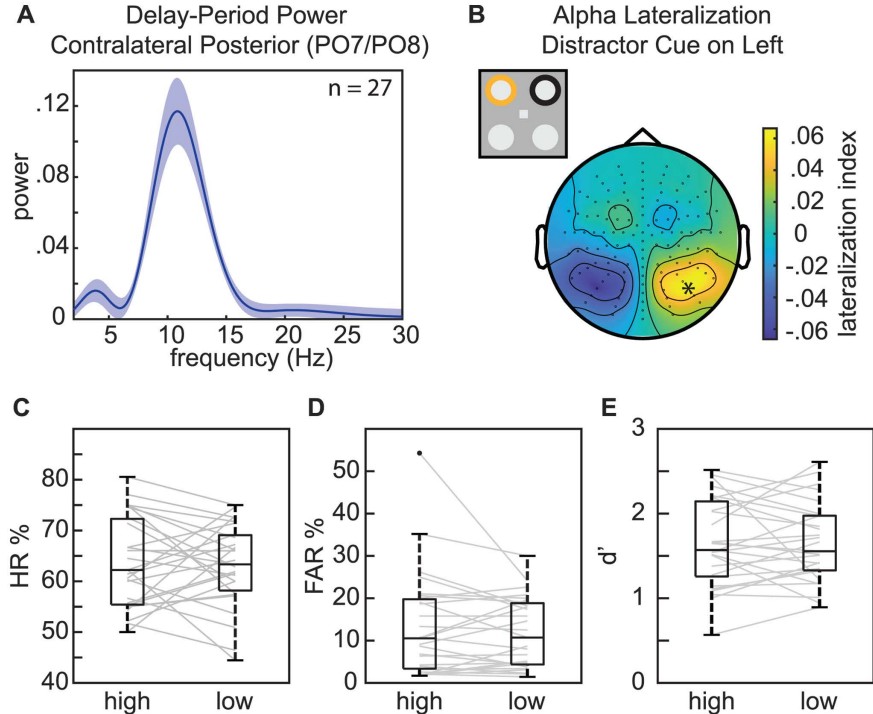

**Fig 10. Posterior alpha power modulation does not affect task performance. A**, Delay-period (−500 to 0 ms relative to stimulus onset) power spectrum for frequencies from 3 to 30 Hz at occipital channels contralateral to the distractor cue. **B**, Topography of alpha power lateralization relative to distractor cues presented on the left side. Channels were flipped across the midline for trials with the distractor cue on the right side. The asterisk indicates the channel used for binning trials by high and low alpha power lateralization. **C**, HR for trials binned by high and low alpha power lateralization using a median split. The line within each box represents the median, while the boundaries of each box represent the upper and lower quartiles, respectively. Lines outside the box represent maximum and minimum data points. Gray lines indicate the change across conditions for individual subjects. **D**, Same as **C**, but for FAR. **E**, Same as **C** but for d′. See S3 Data for individual scores.

for alpha-power lateralization confirmed greater power over occipital electrodes that were contralateral to the distractor cue (Fig 10B), consistent with previous findings [51]; however, alpha-power lateralization at the occipital electrodes with the greatest alpha lateralization (A10/B7) was not associated with behavioral performance on trials with distractors [HR: $t_{(26)}$ = 0.275, $p$ = 0.786, $d$ = 0.058; FAR: $t_{(26)}$ = 0.888, $p$ = 0.383, $d$ = 0.12; d′: $t_{(26)}$ = 0.143, $p$ = 0.888, $d$ = 0.022] (Fig 10C–10E; for individual scores see S3 Data). These results indicate that alpha-power lateralization in the present experiment—unlike alpha phase (see the previous analysis)—was not associated with suppression of distractor processing. As there was no evidence of lateralized phasic modulation within the theta-frequency band, we did not conduct similar tests based on lateralized theta power.

## Discussion

The present findings confirm that rhythmically occurring periods of worse target detection at a to-be-attended location are also associated with greater susceptibility to distractors (i.e., as indicated by higher false alarms), despite the location of those distractors being spatially predictable. These findings are consistent with the Rhythmic Theory of Attention [4], which proposes theta-rhythmic temporal windows associated with a greater likelihood of attentional shifts (i.e., theta-rhythmically occurring "shifting states"). Behavioral performance (i.e., HR and d′) on trials with and without distractors fluctuated as a function of pre-stimulus theta phase at central electrodes, peaking at ~7 Hz. While theta phase was associated with fluctuations in distractor susceptibility, affecting FAR only on trials with a distractor, it was not associated with changes

in distractor processing (i.e., it did not modulate the amplitude of distractor-evoked responses). In addition to theta-dependent effects, behavioral performance on trials with distractors further fluctuated with pre-stimulus alpha phase at both frontocentral and occipital electrodes, peaking at ~9 to 10 Hz. Alpha-band effects (i) emerged exclusively on trials with distractors, (ii) were stronger at occipital electrodes that were contralateral to distractor locations, and (iii) modulated the amplitude of distractor-evoked responses. This pattern of results is consistent with an alpha-mediated gating of distractor processing [30]. The present findings thus confirm a key prediction of the Rhythmic Theory of Attention—rhythmic susceptibility to distracting information—and provide evidence for distinct, frequency-specific attentional mechanisms that phasically modulate the influence of distractors on task performance.

It should be noted that we limited the width of the wavelets used to measure frequency-specific neural activity (i.e., we used a low number of cycles). This was done to obtain phase measurements that were as close as possible to stimulus onset (and subsequent behavioral measures), while also avoiding stimulus-evoked activity. There is therefore considerable spectral smearing across the theta- and alpha-frequency ranges. The present results, however, are consistent with distinct oscillatory mechanisms based on differences in peak frequency, topographical patterns, and functional relationships (e.g., alpha-specific links to distractor processing).

A growing body of literature has demonstrated that spatial attention involves theta-rhythmic processes (for reviews see [4,20,52,53]), revealing both theta-rhythmic modulation of neural responses [6,12] and theta-rhythmic modulation of behavioral performance [5,7–10,54]. That is, there is a "good" theta phase associated with better visual-target detection at the to-be-attended location and a "bad" theta phase associated with worse visual-target detection at the to-be-attended location. Research in non-human primates has specifically linked these theta-rhythmic fluctuations in neural responses and behavioral performance to cortical and subcortical regions within the large-scale network that directs spatial attention, including frontal and parietal cortices [14,15,17,18]. The Rhythmic Theory of Attention [4] proposes that theta-rhythmic fluctuations during attention-related sampling could promote critical cognitive flexibility, suggesting that temporal windows of relatively enhanced processing at the to-be-attended location (i.e., a "sampling state") alternate with temporal windows associated with a higher likelihood of attentional shifts (i.e., a 'shifting state'). This proposal is based on the specific cell types and neural dynamics that characterize alternating, theta-dependent attentional states [4,14]. Although such theta-rhythmically occurring windows of opportunity for shifting attention (i.e., proposed "shifting states") might prevent us from becoming overly focused on any single location in the environment, these temporal windows might also make us more susceptible to salient information that can interfere with task performance (e.g., a distractor at a non-target location). The present results are consistent with this prediction, demonstrating that the theta phase associated with a lower HR (i.e., the proposed "shifting state") was also associated with a greater distractor-related increase in FAR (Fig 8B). We are thus theta-rhythmically more susceptible to distracting information, meaning that windows of opportunity for shifting attentional resources can have behavioral disadvantages—at least in the presence of salient distractors. It should be further noted that theta-rhythmic fluctuations in distractor susceptibility occurred despite the location of the distractors being spatially predictable. That is, mechanisms of attention-related suppression—associated with a spatially informative distractor cue (Figs 3 and 4)—did not overcome the theta-rhythmic tendency to shift attentional resources [4,7,11,19,20].

Previous research has shown that spatially predictable distractors can be actively suppressed to improve target detection elsewhere [43,55–58]. The present experiment replicated such findings, demonstrating that spatially informative distractor cues promoted the suppression of distractors. That is, distractor cues were associated with both better visual-target detection (Fig 3) and lower-amplitude distractor-evoked responses (Fig 4). Previous research suggests that this process of active suppression—like attention-related sampling—might involve a neuro-oscillatory mechanism. Specifically, neural activity in the alpha-frequency band (9–14 Hz) has been repeatedly linked to the suppression of sensory processing [30,31]. For example, alpha power is typically higher in neural populations processing task-irrelevant locations and lower in neural populations processing task-relevant locations [33,34,59,60]. Further supporting this link between higher

alpha power and the suppression of sensory processing, higher alpha power is associated with lower cortical excitability, as measured via spike rates and high-frequency band activity [35–38]. Despite such frequently observed links between alpha-band activity and sensory suppression, results have been mixed regarding whether alpha-mediated mechanisms of spatial attention can be actively deployed to suppress spatially predictable distractors [40,61]. Some studies have reported increased alpha power in anticipation of spatially predictable distractors, with associated effects on behavioral performance [28,51] and distractor-evoked responses [58], while other studies have found no link between alpha power and the suppression of spatially predictable distractors [43,62,63]. The present results replicated an anticipatory lateralization of alpha power relative to the location of target and distractor cues, with higher power at electrodes contralateral to the distractor cue; however, the degree of this cue-related lateralization was not associated with behavioral performance (Fig 10). Here, we instead observed a significant relationship between all behavioral measures (i.e., HR, FAR, and d′) and the pre-stimulus phase of alpha-band activity (Fig 6). Previous research has similarly demonstrated a relationship between pre-stimulus alpha phase and behavioral performance [64–67], but the present results specifically link these effects to distractor processing (also see [68]). That is, alpha-mediated fluctuations in behavioral performance only emerged on trials with distractors and were relatively stronger at occipital electrodes that were contralateral to distractors (Figs 6 and 9). These findings are therefore consistent with an alpha-mediated gating of distractor processing [30] that co-occurs with theta-rhythmic sampling and shifting [4].

Theta- and alpha-related effects in the present experiment were associated with different scalp topographies and different experimental conditions (i.e., alpha-related effects only emerged on trials with distractors). Fig 7 shows the topographies associated with phase–behavior relationships and indicates the electrodes that had significant effects, demonstrating a relationship between theta phase and behavioral performance during trials with and without a distractor (at central electrodes). While comparisons between these conditions using cluster-based permutation tests revealed greater coupling between theta phase (6–7 Hz) and behavioral performance during no-distractor trials (Fig 6B and 6C), topographic comparisons revealed no between-condition differences at theta frequencies (i.e., no differences between no-distractor and distractor trials). In contrast, these same analyses revealed significant between-condition differences at higher frequencies (9 Hz), reflecting alpha-band effects that were exclusive to trials with a distractor (at frontocentral and occipital electrodes). Furthermore, alpha phase—but not theta phase—was associated with fluctuations in distractor-evoked responses (Figs 8D and 9D). The present results thus suggest that alpha-mediated mechanisms of spatial attention specifically influence the visual processing of distractors. Theta-mediated mechanisms of spatial attention, in comparison, might only indirectly influence distractor processing, by modulating the visual processing of targets (i.e., stimuli occurring at cued target locations). During the "shifting state," weaker attention-related enhancement of sensory processing at the to-be-attended location could create a relative advantage for stimuli occurring elsewhere, thereby increasing the likelihood of attentional shifts toward salient distractors. Here, we were unable to test whether the visual response at cued target locations fluctuated as a function of oscillatory phase because the low-contrast targets in the present experiment did not elicit a clear evoked response (S1 Fig). The observed differences associated with theta- and alpha-mediated effects, however, are consistent with previous work that has proposed different functional roles for attention-related theta- and alpha-band activity, with theta-band activity perhaps being more closely associated with attention-related sampling and exploration and alpha-band activity perhaps being more closely associated with sensory processing and suppression [54,69–72].

The present findings (and those of others) provide evidence for theta- and alpha-mediated mechanisms of spatial attention, but it remains largely unknown whether and how these mechanisms interact. Several previous studies have described theta-rhythmic modulations of alpha power [9,14,15,73], with such interactions possibly being mediated through feedback projections from higher-order regions to visual cortex [74–79]. Helfrich and colleagues [78], for example, demonstrated that the phase of low-frequency oscillations in prefrontal cortex (~2 to 4 Hz) modulated the effect of posterior alpha-band activity on behavioral performance. Here, we observed theta- and alpha-related effects with close proximity

in the frequency domain (i.e., these effects occurred at similar peak frequencies), making it more difficult to conceive of a nested relationship, whereby theta phase modulates alpha power [80,81]. It might be possible, however, that theta-band activity in higher-order regions (e.g., frontal cortex) influences alpha phase in visual cortex [69]—rather than alpha power—optimally aligning theta- and alpha-mediated mechanisms of spatial attention. Future research in non-human primates, which can measure interactions between different neural populations, will need to determine whether there are interactions between theta- and alpha-mediated effects on distractor processing.

The present results confirm that theta-rhythmic, attention-related sampling is associated with temporal windows of greater distractor susceptibility. That is, windows of opportunity for shifting attentional resources from the present focus of attention to another location increase the likelihood that a salient distractor will capture attentional resources. Rhythmic attention-related sampling—while perhaps preventing us from becoming overly focused on any single location—can also lead to behavioral disadvantages (i.e., increased distractor effects). These theta-rhythmically occurring fluctuations in distractor susceptibility co-occurred with alpha-mediated effects on distractor processing. Alpha-band activity contralateral to cued distractor locations phasically modulated both behavioral measures and distractor-evoked visual responses, consistent with an alpha-mediated gating of distractor processing. The present findings thus provide evidence for distinct theta- and alpha-mediated mechanisms of spatial attention, with both mechanisms phasically modulating the extent to which spatially predictable, high-contrast distractors interfere with the detection of spatially predictable, low-contrast visual targets.

## Materials and methods

### Ethics statement

All study procedures were conducted in accordance with the Declaration of Helsinki following protocols approved by the Research Subjects Review Board at the University of Rochester (ID: STUDY00005678).

### Subjects

Forty individuals with normal or corrected-to-normal vision and no history of neurological disorders participated in the experiment (26 females, 14 males; mean age, 24.1 years). All subjects provided informed consent in writing.

### Apparatus and stimuli

Subjects were seated in a comfortable chair in a sound- and light-attenuated chamber. Chair and table heights were adjusted so that the subject could comfortably use padded chin and forehead rests. A foot stool was provided if needed. Subjects were instructed to remain as relaxed as possible during trials to minimize noise in the EEG attributable to muscle activity. Task contingencies were controlled with custom Presentation (Neurobehavioral Systems) scripts run on a Dell Precision 5820 desktop computer with a 27-inch Acer Predator XB2 LCD monitor (1,920 × 1,080 pixels at 240 Hz) positioned 57 cm from the subject's eyes. Fig 1 depicts the primary experimental display which consisted of a gray background with a light gray fixation square (0.5 × 0.5 dva) at the center of the screen and four light-gray placeholder circles of 4 dva centered at 6 dva from fixation and 45°, 135°, 225°, and 315° from vertical (i.e., one in each quadrant). Cues consisted of a solid border (outside diameter of 4.5 dva) around placeholder stimuli. Target cues were dark gray and distractor cues were orange or blue. Targets were near-threshold, gray, circular patches, 4 dva in diameter. Target luminance was adjusted throughout the task using single integer adjustments to RGB values if hit rate (HR) fell below 50% or rose above 70% for a series of 10 valid-target trials. Target luminance was also adjusted up if false alarm rate (FAR) rose above 30% for a sequence of 10 trials with a false alarm or correct rejection at the cued target position. Due to random interleaving of trials, target luminance was independent of any experimental variables (e.g., distractor presence). Distractors were highly salient, circular patches, 6 dva in diameter, and drawn in orange or blue (same colors as distractor cues).

## Behavioral task

Fig 1 illustrates the task. Participants pressed the left button on a computer mouse to begin each trial, causing a fixation square to appear for 500–1,000 ms. Four placeholders then appeared for 500–1,000 ms. Next, subjects were presented with a cue display consisting of a target cue (black) at one of the two positions on either side of the visual field and a distractor cue (randomly orange or blue) at one of the two positions on the opposite side. Cues remained on the screen for 100 ms. The subsequent display consisted of the four placeholder stimuli and lasted between 500 and 1,600 ms, with the duration selected randomly from a uniform distribution. After the delay period, stimuli were presented for 100 ms. The stimulus configuration could consist of either i) four placeholder stimuli (i.e., catch trials; ~17.6% of trials), ii) a target stimulus and three placeholder stimuli (~23.5%), iii) a distractor stimulus and three placeholder stimuli (~23.5%), or iv) a target, a distractor, and two placeholder stimuli (~35.3%). Seventy percent of all targets/distractors appeared at cued locations (i.e., valid), while 30% of targets/distractors appeared at non-cued locations (i.e., invalid). We included a small percentage of invalidly cued trials (i.e., 30%) to confirm that participants were selectively deploying spatial attention based on the spatially informative cues (Figs 3 and 4). As with our previous work [14,16], the necessarily low percentage of invalid trials (i.e., low trial counts) precluded us from analyzing phase–behavior relationships associated with invalidly cued trials. In contrast to valid target/distractor conditions which included 180 trials prior to additional exclusions (e.g., minimum delay duration), each subject performed only 60 trials of each invalid condition. Distractor color (orange or blue) was randomly selected for each trial and independent of cue color. Following the offset of stimuli, placeholders returned to the screen for a brief period (300 ms). Participants were then shown a response screen with instructions to press 1 on the keyboard if they detected a target at the cued location, press 2 if they detected a target at a non-cued location, or press 3 if they did not detect a target. This display remained until participants responded.

Each subject completed at least 20 practice trials followed by six blocks of 170 trials for a total of 1,020 trials. Subjects were instructed to maintain fixation without blinking for the duration of each trial (i.e., the period beginning with the subject pressing the mouse button and ending with onset of the response screen). Fixation was monitored using an infrared eye-tracking camera (EyeLink 1000 Plus, SR Research). If fixation was broken (>2 dva), the trial was aborted, and a warning message appeared, instructing subjects to begin the next trial. If trials were aborted in this way, additional trials were added to the block to ensure the total number of completed trials for each condition was consistent across subjects.

## EEG recording and pre-processing

EEG data were collected using a BioSemi ActiveTwo system (BioSemi) with 128 active Ag/AgCl electrodes. Data were digitized at 2,048 Hz and downsampled to 512 Hz off-line. All electrode impedances were kept below 20 kΩ. To ensure consistent placement of the EEG cap, the vertex electrode (A1) was placed at 50% of the distance between the inion and the nasion and between the tragus on the left and the right ears. EEG data were analyzed using the FieldTrip toolbox [82] and custom scripts written in MATLAB (R2023a). Off-line, continuous EEG data were re-referenced to the average of all 128 channels, high-pass filtered (fourth-order Butterworth with 0.1 Hz cutoff), and then segmented into epochs relative to stimulus onset (−2 to 0.5 s). Bad channels were visually identified during recording sessions. Artifacts related to blinks and eye movements were excluded from the data by design, as task performance was contingent on maintaining fixation (i.e., trials with a blink or saccade were aborted). Data were visually inspected and any trials exceeding a ±100 μV threshold during analysis periods (e.g., delay period or ERP window) were interpolated using a distance-weighted average of the four nearest good electrodes, if the average distance was <5 cm (otherwise data were excluded). Trials without all 128 electrodes were excluded from analyses and subjects with greater than 50% of these trials were excluded entirely, resulting in 8 subjects being excluded (*n* = 32).

## Behavioral analysis

Based on visual inspection of response time histograms for individual subjects, trials with a response time less than 400 ms or greater than 2000 ms were excluded from these analyses. Paired samples t-tests were used to test for effects of target cueing on HR, FAR, and d-prime (d′). To isolate target cueing effects, only trials with no distractor were included. For calculations of FAR, trial counts were adjusted to account for the relative probability of targets at cued and non-cued locations (75% versus 25%). Paired samples t-tests were also used to test for effects of distractor presence and distractor cueing on HR, FAR, and d′. To isolate distractor and distractor cueing effects, only trials with valid targets were included in calculations of HR and only trials with responses at cued target locations were included in calculations of FAR. Cohen's d was used to express effect sizes for all behavioral tests.

## ERP analyses

ERP analyses were limited to the time window from −100 to 350 ms relative to distractor onset. This time window captures the sensory response to distractors without including activity evoked by response screen onset. Time series were baseline corrected using the period from −100 to 0 ms and averaged across trials. Only trials without targets were included in these analyses. Trials were randomly subsampled to match counts across conditions for each subject. All analyses used occipital channels A10/B7 (corresponding to PO7/PO8), which were selected a priori based on past work on visual responses to distractors [42–44], as well as past work on other attention-related effects associated with visual responses, such as the N2pc component [45]. For analyses measuring the effects of pre-stimulus phase (see below) on distractor-evoked visual responses, we applied a Hilbert transform to the broadband filtered signal (1–55 Hz) and took the absolute value to capture non-phase-aligned, broad-band power. This procedure allowed us to baseline trials, which could not be done confidently for phase-aligned signals, as pre-stimulus binning based on phase creates systematic differences in the baseline of raw voltage signals. Cluster-based permutation tests were performed to compare distractor-evoked responses across conditions while controlling for multiple comparisons [83]. Briefly, a $t$ test was performed to compare responses at each time point. We then calculated the sum of differences for clusters of consecutive timepoints at which a significant difference was observed ($\alpha = 0.05$). Next, we shuffled the condition assignment across trials and repeated this procedure to identify the maximum sum of differences for a representative null cluster. This procedure was repeated 1,000 times to form a null distribution against which we compared the sum of differences in observed clusters of significant timepoints. Observed clusters with a sum greater than 95% of null cluster sums were considered significant. We also compared ERP difference waves (good phase versus bad phase) to test for differences in the effects of theta and alpha oscillations on distractor processing. For each of 1,000 permutations, maximum cluster size was determined from a final difference wave created by randomly assigning trials to one of the two conditions (theta or alpha), separately for each subject. Observed clusters with a sum greater than 95% of null cluster sums were considered significant.

## Lateralized phase–behavior analyses

Trials with a response time less than 400 ms or greater than 2,000 ms were excluded from these analyses. For each condition (no distractor and valid distractor), trials were binned based on frequency-specific, pre-stimulus phase measurements. Phase was measured on each trial by taking the angle of the complex output produced by convolving the signal with a single frequency-specific Morlet wavelet (3–55 Hz), with a varying number of cycles (2 cycles for 3–8 Hz, increasing logarithmically from 2 to 5 cycles for 9–55 Hz). Subsequent analyses focused on frequencies from 3 to 25 Hz. We only included trials with a delay period of at least 1,000 ms to avoid capturing cue-evoked activity. Wavelets were fit for each frequency at a time point that positioned the last point of the wavelet immediately prior to stimulus onset, to avoid capturing stimulus-evoked activity. Performance measures were calculated for overlapping phase bins with a width of 120° (e.g., 0°–120°) that were shifted forward in 10° steps (e.g., 10°–130°, 20°–140°, etc.). In contrast to prior work [8,14,84],

we leveraged the opposition of target and distractor cues to test for lateralized oscillations with functional links to target detection. In other words, we always measured phases with respect to the positions of relevant cues. This was achieved by flipping channels for trials with the distractor cue appearing on the right side, so that channels on the right side are always contralateral to the distractor cue (and ipsilateral to the target cue). This procedure was repeated to generate phase–behavior functions, spanning all phases, for each frequency-by-channel pair, which were then averaged across participants. Here, we predicted that the averaged phase–behavior functions would have a characteristic shape, with a peak separated from a trough by approximately 180° (i.e., approximating a one-cycle sine wave) (see Fig 5). Following this prediction, a discrete Fourier transform was applied to the phase–behavior function for each frequency-by-channel pair, and the second component—representing a one-cycle sine wave—was kept. The amplitude of this component reflects both how closely the function approximated a one-cycle sine wave and the effect size, thereby providing a single value representing the strength of the phase–behavior relationship [8,14,84]. Phase–behavior analyses were applied to trials with (i) valid targets/responses at cued locations and (ii) either no distractor or a valid distractor. While there are alternative measures of phase–behavior coupling [85], these approaches can suffer from reduced power when trial counts are relatively low (<200) and unbalanced across conditions (e.g., correct and incorrect trials).

Because previous work has also linked the likelihood of microsaccades with theta phase [22], we tested whether excluding trials with microsaccades affected the general pattern of phase–behavior relationships [14]. Saccades were detected using the EyeLink default speed threshold of 30°/s and acceleration threshold of 8,000°/s$^2$. We specifically measured the occurrence of microsaccades (<2 dva) during the window of the longest wavelet used for phase estimates in the present analyses (i.e., −666 to 0 ms relative to stimulus onset). Across subjects, microsaccades occurred on 7.51% of all trials (SD = 0.055).

We used cluster-based permutation tests to determine the statistical significance of observed phase–behavior relationships, while correcting for multiple comparisons [86]. First, we used permutation tests to identify frequency × channel pairs with significant phase–behavior relationships. This was done by comparing each observed phase–behavior value (i.e., one-cycle sine fit)—averaged across subjects—with null values obtained by randomly shuffling phases across trials. After shuffling, we recalculated phase–behavior values for each subject and then averaged across subjects. These steps were repeated 1,000 times to generate a null distribution. Phase–behavior relationships were considered significant for frequency × channel pairs that exceeded 95% of the null values. Next, we calculated the sum of phase–behavior values in clusters of significant frequency × channel pairs that were neighbors in either frequency or space (i.e., we calculated cluster sums). We then calculated the maximum sum of phase–behavior values in clusters obtained from the null datasets (i.e., the datasets obtained from randomly shuffling phases across trials). For each of the 1,000 sets of null phase–behavior functions, we identified frequency × channel pairs with "significant" phase–behavior relationships. As described above for the observed data, null phase–behavior relationships that exceeded 95% of the remaining 999 null values for that frequency x channel pair were considered significant. Null cluster sums were then calculated across frequency and space, and the maximum cluster sum was kept (i.e., we kept the largest cluster sum from each iteration). This was repeated for the remaining 999 null datasets to generate a distribution of maximum cluster sums. Observed cluster sums that exceeded 95% of these null cluster sums were considered statistically significant.

We also adapted this cluster-based permutation approach to test for differences in the spectral and topographic characteristics of phase–behavior effects observed in different conditions (distractor versus no distractor). We calculated observed differences by subtracting the strength of the phase–behavior relationships (i.e., one-cycle sine fits) for the no-distractor condition from the corresponding values for the distractor condition. We then obtained null distributions by (i) randomly shuffling trials across the conditions (i.e., across the no-distractor and distractor conditions) for each subject, (ii) recalculating phase–behavior relationships for the shuffled conditions, (iii) averaging phase–behavior relationships across subjects, and (iv) recalculating the between-condition differences. Frequency x channel pairs with an observed absolute value (i.e., after subtracting strength of phase–behavior relationships for the no-distractor condition from strength

of phase–behavior relationships for the distractor condition) greater than 95% of the absolute values of corresponding null pairs were considered significantly different. The calculation of cluster sums and the determination of statistically significant clusters proceeded as described above (i.e., for the analyses of within-condition phase–behavior relationships).

### Topographic analyses

To compare the topographies of phase–behavior effects, we adapted an approach used to test for differences between ERP topographies [50]. For each condition, the strength of phase–behavior functions (i.e., one-cycle sine fit) was z-normalized across channels. We then calculated the square root of the mean of squared differences between the normalized values to obtain a single value representing the global dissimilarity (DISS) between the topographies for the two conditions [87]. Condition labels were then shuffled at the subject level and global dissimilarity was re-calculated. This process was repeated 1,000 times to create a reference distribution of global dissimilarity values. The p-value for each test was the percentage of global dissimilarity values in the null distribution that exceeded the observed value. We used Bonferroni corrections to account for multiple comparisons ($\alpha = 0.01$).

### Analyses of alpha power

Only trials with delay periods longer than 900 ms were included in analyses of delay-period alpha power to avoid effects from cue-evoked visual responses. Alpha frequency can vary substantially between individuals [88]; therefore, peak alpha frequencies were determined for each subject. Identification of subject-specific alpha peak frequencies used data from channels A10/B7 which correspond to those used in prior work on the role of posterior alpha in attentional selection [43,59,89]. Data were isolated from −500 to 0 ms relative to stimulus onset, and irregular resampling autospectral analysis [90] was used to obtain oscillatory power spectra (2–30 Hz) at electrodes contralateral to cued distractor locations. Data were padded with zeros to obtain a frequency resolution of 0.1 Hz. For each subject, local maxima in the alpha-band (8–15 Hz) were identified after averaging spectra across trials. Subject-specific alpha frequency was used for subsequent analyses of alpha power. For each trial, power was measured during the period from −500 to 0 ms relative to stimulus onset by applying a Hanning taper and discrete Fourier transform to the signal and taking the squared magnitude of the complex output. Data were padded with zeros to achieve 0.1 Hz resolution. Pre-stimulus alpha power was used to calculate a lateralization index using the following formula:

$$\text{LI}_{\text{contra–ipsi}} = (\text{Power}_{\text{contra}} - \text{Power}_{\text{ipsi}})/(\text{Power}_{\text{contra}} + \text{Power}_{\text{ipsi}})$$

For trials with a distractor cue on the right side, lateralization indices for all channels were flipped across the midline before combining with data for trials with the distractor cue on the left side. Trials were binned by alpha lateralization (median split) at the occipital channel (A10/B7) contralateral to distractor cues and paired samples t-tests were used to test for an effect on task performance.

### Supporting information

**S1 Fig. Target-evoked responses (i.e., associated with a low-contrast target) and distractor-evoked responses (i.e., associated with a salient distractor). A**, Grand average ERPs showing responses to targets without distractors (blue) and distractors without targets (red). **B**, As in **A**, but depicting the broadband (1–55 Hz) responses to the same stimuli. Error bars indicate ±1 SEM.
(EPS)

**S2 Fig. Alpha phase–behavior effects for posterior channels. A**, Phase-HR (solid blue line) and phase-FAR (solid red line) relationships for 9-Hz phase using no-distractor trials. These effects were averaged across posterior channels that

were contralateral to the distractor cue and significant for trials with a valid distractor (depicted in inset). **B**, Same as **A**, but for trials with a valid distractor. All error bars indicate ±1 SEM.
(EPS)

**S3 Fig. Phase-d′ effects for all trials and trials without microsaccades. A**, Plots depicting the strength of phase-d′ relationships as a function of frequency (3–25 Hz) across all 128 channels. **B**, As in **A**, but for only trials without microsaccades (< 2 dva).
(EPS)

**S4 Fig. Comparison between phasic modulation of distractor-evoked responses at 7 and 9 Hz. A**, Difference waves calculated from distractor-evoked broadband (1–55 Hz) responses as a function of pre-stimulus phase ("good" vs. "bad") for 7 Hz (blue line) and 9 Hz (red line). Error bars indicate ±1 SEM. Horizontal blue and red bar indicates the time window of significant differences between conditions (159.9–206.8 ms). **B**, Topography of the difference wave for the 9 Hz effect averaged over the significant time window (163.8–218.5 ms). This plot is configured as if all stimuli appeared on the left side of the visual field (i.e., channels were flipped for trials with stimulus on the right). The electrode used for ERP analyses—A10 or B7 (corresponding to PO7 and PO8), depending on whether the distractor was presented to the left or right side of the visual field—is depicted with a bold dot.
(EPS)

**S1 Data. Distractors impaired task performance.** Individual scores (*n* = 32) for analyses of distractor effects on HR (Fig 2A), FAR (Fig 2B), and d′ (Fig 2C).
(XLSX)

**S2 Data. Spatially informative target cues and distractor cues improved task performance.** Individual scores (*n* = 32) for analyses of target cueing effects on HR (Fig 3A), FAR (Fig 3B), and d′ (Fig 3C) and analyses of distractor cueing effects on HR (Fig 3D), FAR (Fig 3E), and d′ (Fig 3F).
(XLSX)

**S3 Data. Posterior alpha power modulation does not affect task performance.** Individual scores (*n* = 27) for analyses of alpha power lateralization on HR (Fig 10C), FAR (Fig 10D), and d′ (Fig 10E).
(XLSX)

## Acknowledgments

We would like to thank Miral Abdalaziz for assistance during data collection and Edmund Lalor for valuable advice regarding topographic analyses.

## Author contributions

**Conceptualization:** Ian C. Fiebelkorn.

**Data curation:** Zach V. Redding.

**Formal analysis:** Zach V. Redding, Yun Ding.

**Funding acquisition:** Ian C. Fiebelkorn.

**Investigation:** Zach V. Redding.

**Methodology:** Zach V. Redding, Ian C. Fiebelkorn.

**Project administration:** Zach V. Redding, Ian C. Fiebelkorn.

**Resources:** Ian C. Fiebelkorn.

**Software:** Ian C. Fiebelkorn.

**Supervision:** Ian C. Fiebelkorn.

**Validation:** Zach V. Redding.

**Visualization:** Zach V. Redding.

**Writing – original draft:** Zach V. Redding.

**Writing – review & editing:** Zach V. Redding, Yun Ding, Ian C. Fiebelkorn.

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
