## [Editor Report · Decision Letter 0]

17 Dec 2025

Dear Ian,

Thank you for submitting your revised manuscript entitled "Attention-related sampling of targets rhythmically alternates with increased susceptibility to co-occurring distractors" for consideration as a Research Article by PLOS Biology.

Your manuscript has now been evaluated by the PLOS Biology editorial staff as well as by an academic editor with relevant expertise and I am writing to let you know that we would like to send your submission back to the reviewers.

Once your full submission is complete, your paper will undergo a series of checks in preparation for peer review. After your manuscript has passed the checks it will be sent out for review. To provide the metadata for your submission, please Login to Editorial Manager (https://www.editorialmanager.com/pbiology) within two working days, i.e. by Dec 19 2025 11:59PM.

Kind regards,

Christian

Christian Schnell, PhD

Senior Editor

PLOS Biology

cschnell@plos.org

---

## [Decision Letter · Decision Letter 1]

21 Jan 2026

Dear Ian,

Thank you for your patience while we considered your revised manuscript "Attention-related sampling of targets rhythmically alternates with increased susceptibility to co-occurring distractors" for consideration as a Research Article at PLOS Biology. Your revised study has now been evaluated by the PLOS Biology editors, the Academic Editor and two of the original reviewers.

In light of the reviews, which you will find at the end of this email, we are pleased to offer you the opportunity to address the remaining points from the reviewers in a revision that we anticipate should not take you very long. We will then assess your revised manuscript and your response to the reviewers' comments with our Academic Editor aiming to avoid further rounds of peer-review, although we might need to consult with the reviewers, depending on the nature of the revisions. Please let us know if you need more time than the 30 days, I'd be happy to extend the deadline if needed.

Please also make sure to address the following data and other policy-related requests:

* We would like to suggest a different title to improve its accessibility for our broad audience:

Frequency-specific attentional mechanisms phasically modulate the influence of distractors on task performance

* Please add the links to the funding agencies in the Financial Disclosure statement in the manuscript details.

* Please include the approval/license number of the ethical approval for the experiments.

* Please include information in the Methods section whether the study has been conducted according to the principles expressed in the Declaration of Helsinki.

* Please specify whether the participants provided written or oral consent.

* DATA POLICY:

Regardless of the method selected, please ensure that you provide the individual numerical values that underlie the summary data displayed in the following figure panels as they are essential for readers to assess your analysis and to reproduce it: 2ABC, 3ABCDEF and 10CDE.

* CODE POLICY

**IMPORTANT - SUBMITTING YOUR REVISION**

*Resubmission Checklist*

*Published Peer Review*

*PLOS Data Policy*

*Blot and Gel Data Policy*

Sincerely,

Christian

Christian Schnell, PhD

Senior Editor

PLOS Biology

cschnell@plos.org

REVIEWS:

Reviewer #1 (signed as Biao Han): The authors have addressed the main methodological concerns raised in my initial review. The revised manuscript now clarifies several critical analysis choices, including the a priori rationale for ERP electrode selection, the temporal definition of pre-stimulus phase estimation, and the statistical framework used to evaluate phase-behavior relationships. The inclusion of cluster-based permutation procedures and additional control analyses improves the robustness and transparency of the reported effects.

The authors also provide formal statistical comparisons supporting frequency-specific differences between theta- and alpha-band effects, which strengthens the interpretability of the central claims. While some analyses could in principle be extended further, the current revisions are sufficient to support the conclusions as stated.

Reviewer #2: The authors have done a good job of addressing all my comments, except for my first point. Apparently, I did not formulate it clearly enough. My point was that it would be desirable for the authors to provide an analysis of behavioral oscillations, such as in Figure 2 in Fiebelkorn et al. 2018 and Figure 1 in Helfrich et al, 2018. Such time-resolved behavioral time courses would provide a solid functional basis for the fact that distractor susceptibility fluctuates over time.

Reviewer #3 had no further comments.

---

## [Editor Report · Decision Letter 2]

4 Feb 2026

Dear Ian,

Thank you for the submission of your revised Research Article "Frequency-specific attentional mechanisms phasically modulate the influence of distractors on task performance" for publication in PLOS Biology. On behalf of my colleagues and the Academic Editor, Huan Luo, I am pleased to say that we can in principle accept your manuscript for publication, provided you address any remaining formatting and reporting issues. These will be detailed in an email you should receive within 2-3 business days from our colleagues in the journal operations team; no action is required from you until then. Please note that we will not be able to formally accept your manuscript and schedule it for publication until you have completed any requested changes.

PRESS

Sincerely,

Christian

Christian Schnell, PhD

Senior Editor

PLOS Biology

cschnell@plos.org